

# Development and validation of a nomogram to predict postoperative pulmonary complications following thoracoscopic surgery

Bin Wang[*], Zhenxing Chen[*], Ru Zhao, Li Zhang and Ye Zhang

Department of Anesthesiology and Perioperative Medicine, The Second Affiliated Hospital of Anhui Medical University, Hefei, China

[*] These authors contributed equally to this work.

## ABSTRACT

**Background.** Postoperative pulmonary complications (PPCs) after thoracoscopic surgery are common. This retrospective study aimed to develop a nomogram to predict PPCs in thoracoscopic surgery.

**Methods.** A total of 905 patients who underwent thoracoscopy were randomly enrolled and divided into a training cohort and a validation cohort at 80%:20%. The training cohort was used to develop a nomogram model, and the validation cohort was used to validate the model. Univariate and multivariable logistic regression were applied to screen risk factors for PPCs, and the nomogram was incorporated in the training cohort. The discriminative ability and calibration of the nomogram for predicting PPCs were assessed using C-indices and calibration plots.

**Results.** Among the patients, 207 (22.87%) presented PPCs, including 166 cases in the training cohort and 41 cases in the validation cohort. Using backward stepwise selection of clinically important variables with the Akaike information criterion (AIC) in the training cohort, the following seven variables were incorporated for predicting PPCs: American Society of Anesthesiologists (ASA) grade III/IV, operation time longer than 180 min, one-lung ventilation time longer than 60 min, and history of stroke, heart disease, chronic obstructive pulmonary disease (COPD) and smoking. With incorporation of these factors, the nomogram achieved good C-indices of 0.894 (95% confidence interval (CI) [0.866–0.921]) and 0.868 (95% CI [0.811–0.925]) in the training and validation cohorts, respectively, with well-fitted calibration curves.

**Conclusion.** The nomogram offers good predictive performance for PPCs after thoracoscopic surgery. This model may help distinguish the risk of PPCs and make reasonable treatment choices.

# BACKGROUND

Postoperative pulmonary complications (PPCs) are very common in thoracic surgery and are major causes of increased morbidity, mortality, and prolonged hospital stay during the perioperative period (*Ball et al., 2018*; *Haller & Walder, 2017*; *Hayami et al., 2018*; *Li et*

Corresponding author
Ye Zhang, zhangye_hassan@sina.com

*al., 2020*; *Mathis et al., 2019*; *Moore et al., 2017*; *Sayal et al., 2018*). Thoracic surgery tends to involve one-lung ventilation, long operation times, increased trauma, sternum or rib incision, nerve injury, and acute or chronic pain. As these characteristics lead to many complications and seriously affect the prognosis of patients, PPCs of thoracic surgery have been widely studied by surgeons and anesthesiologists (*Hoshikawa & Tochii, 2017*; *Khaitan & D'Amico, 2018*; *Medbery & Force, 2017*; *Munoz De Cabo et al., 2020*). The incidence of PPCs after thoracic surgery is 19–59% (*Agostini et al., 2010*; *Garcia-Miguel, Serrano-Aguilar & Lopez-Bastida, 2003*).

Thoracoscopic surgery reduces the risk of PPCs (*Cypel & Yasufuku, 2018*; *Wong, Sit & Au, 2018*). Some studies have found that compared to thoracotomy, thoracoscopic surgery significantly reduces the incidence of PPCs. Nevertheless, the incidence of PPCs after thoracoscopic surgery is still as high as 16–30% due to the characteristics of the surgery, such as the need for one-lung ventilation and lung or diaphragm surgery (*Kaufmann et al., 2019*; *Nozaki et al., 2018*).

The most common risk factors for pulmonary complications in thoracic surgery include older age, cardiovascular complications, smoking, and chronic obstructive pulmonary disease (COPD) (*Licker et al., 2006*; *Stephan et al., 2000*). Although independent risk factors for PPCs in thoracic surgery have been examined, a simple and effective prediction model has not yet been established, and specific predictive studies on PPCs after thoracoscopic surgery have not been carried out.

The present study aimed to identify risk factors for PPCs of thoracoscopic surgery based on demographic and clinical data of patients and to establish a nomogram for predicting PPCs through independent risk factors that could serve as a reference for clinical practice.

## METHODS

### Study population

This retrospective study was approved by the Ethics Committee of the Second Affiliated Hospital of Anhui Medical University (No. YX2020-038) and registered in the Chinese Clinical Trial Registry (No. ChiCTR2000033329), in accordance with the TRIPOD checklist. From August 2018 to March 2020, 905 patients who underwent thoracoscopic surgery at the Second Affiliated Hospital of Anhui Medical University were enrolled and analyzed. The following were exclusion criteria: emergency and trauma patients; age <18 years old; patients with pulmonary infection and pleural effusion before the operation; patients who underwent thoracotomy or if the operation was cancelled; patients who underwent re-operations; and patients whose relevant data were missing or incomplete.

According to the requirements of the ethics committee, informed consent was carried out according to the following methods: A. If the patient has passed away or cannot be contacted, apply for exemption of informed consent according to the process of ethics committee; B. If patients agree to include their medical records in this study, they should sign the informed consent; C. If the patient agrees to provide his/her data for this study while does not want to sign the informed consent form, a witness must be present. The data can only be included after the witness signed in the informed consent; D. If patients explicitly refuse to include their medical records in this study, their data will be excluded.

## Data collection

Based on clinical importance, scientific knowledge, previous research (*Agostini et al., 2010*; *Licker et al., 2006*; *Stephan et al., 2000*) and the actual data that could be collected for a retrospective study, 27 demographic and clinical variables of patients undergoing thoracoscopic surgery were collected. General information included age, sex, American Society of Anesthesiologists (ASA) grade, preoperative mean arterial pressure (MAP), body mass index (BMI), and history of hypertension, diabetes mellitus, stroke, heart disease, COPD, smoking (ever or current), and alcohol use. Laboratory examination data before the operation included leukocyte, red blood cell, and platelet counts and aspartate aminotransferase, alanine aminotransferase, blood glucose, serum creatinine, blood urea nitrogen, serum sodium, serum potassium, and serum albumin levels. Information on the operation and anesthesia management included the duration of the operation, airway management, the duration of one-lung ventilation, and tpyes of operation.

The primary end-point of this study was PPCs, which were defined according to the European Perioperative Clinical Outcome (EPCO), including respiratory infection, respiratory failure, pleural effusion, atelectasis, pneumothorax, bronchospasm, and aspiration pneumonia (*Jammer et al., 2015*; *Miskovic & Lumb, 2017*) (Table S1). An event was considered a PPC if one of these complications was detected within the first 7 postoperative days. The diagnosis of PPCs was made by two experienced researchers based on the patient's clinical data. In the case of a dispute, the diagnosis was discussed by the research group.

## Statistical analysis

All patient demographic and clinical data were randomly divided into a training cohort and validation cohort according to an 80%:20% ratio by setting the random seed "1234567" in the "caTools" package of R software (version 3.6.3; http://www.r-project.org). The training and validation cohorts were used to derive and validate the model, respectively. Continuous variables such as age and BMI were categorized after being assessed based on clinical reference values and scientific knowledge. The data were compared using the $\chi^2$ test or Fisher's exact test, and the results are expressed as whole numbers and proportions.

Univariate logistic regression analysis was performed to identify risk factors for PPCs following thoracoscopic surgery in the training cohort. All variables associated with PPCs with a *p*-value less than 0.1 were candidates for backward stepwise multivariate analysis with the Akaike information criterion (AIC) to investigate independent risk factors. The nomogram was formulated based on the results of AIC multivariate logistic regression analysis in the training cohort, and selected variables were incorporated in the nomogram to predict the probability of PPCs.

The total scores of the nomogram were calculated for each patient in the training cohort. A receiver operating characteristic (ROC) curve was analyzed to calculate the optimal cut-off points based on the Youden index in the training cohort, after which the cut-off points in the training cohort were used in both cohorts to verify the generalizability of the model. The accuracy, precision, sensitivity, specificity, and the concordance index (C-index) were

calculated, and ROC curves were constructed for the training and validation cohorts. The predictive performance of the nomogram was assessed by the C-index with 1000 bootstrap replicates and calibrated with 100 bootstrap replicates to decrease overfitting in the training and validation cohorts, respectively. A C-index value of 0.5 indicates no discrimination; a value of 1.0 indicates perfect discrimination ability with different outcomes. In an excellently calibrated model, the predictions should fall on a diagonal line of 45 degrees, which represents the ideal relationship between the observed outcome frequencies and the predicted probabilities.

All tests were 2-tailed, and $p$-values less than 0.05 were considered statistically significant. Statistical analysis was carried out using SPSS 22.0 software (Chicago, IL, USA) and R software. The R software packages included rms, foreign, caTools, tableone and pROC.

## RESULTS

### Patient characteristics

During the study period, 1084 consecutive patients underwent thoracoscopic surgery. Among the cases, 48 involved emergency and trauma surgery, 13 patients were younger than 18 years old, 21 cases were converted to open surgery, and there were missing or incomplete data for 97 patients. No patient explicitly refused to include his/her data for this research. Thus, a total of 905 patients who met the inclusion criteria were included in the present study. All patient demographic and clinical data were randomly divided into a training cohort (724 cases) and a validation cohort (181 cases) at a ratio of 80%:20% (Fig. S1). Of the included cases, 207 patients exhibited PPCs, with an incidence of 22.87%. The details of the PPCs are shown in Table S2. Among those who experienced PPCs, 166 patients (22.93%) were allocated to the training cohort, and 41 patients (22.65%) were allotted to the validation cohort. The demographic and clinical data of the training and validation cohorts are shown in Table 1. The $p$-values and standardized mean differences (SMDs) suggested that the partition of training cohort data and validation cohort data was balanced.

### Independent risk factors for PPCs

Backward stepwise selection using the AIC in logistic regression modelling identified the following 7 variables as having the strongest associations with PPC risk: ASA grade III/IV, operation time longer than 180 min, one-lung ventilation time longer than 60 min, and history of stroke, heart disease, COPD, and smoking. According to the multivariable analysis, ASA grade III/IV [odds ratio (OR), 10.171; 95% confidence interval (CI) [5.598–18.926]; $p < 0.001$], operation time longer than 180 min (OR, 2.480; 95% CI [1.363–4.563]; $p = 0.003$) or longer than 300 min (OR, 12.516; 95% CI [5.232–17.766]; $p < 0.001$), one-lung ventilation time longer than 60 min (OR, 2.947; 95% CI [1.257–7.648]; $p = 0.018$), history of stroke (OR, 4.209; 95% CI [1.518–11.683]; $p = 0.006$), history of COPD (OR, 2.888; 95% CI [1.230–6.980]; $p = 0.016$), and history of smoking (OR, 4.101; 95% CI [2.544–6.654]; $p < 0.001$) were each independently associated with PPCs (Table 2).

**Table 1  Descriptive variables of the patients following thoracoscopic surgery.**

| | Total (n = 905) | Training cohort (n = 724) | Validation cohort (n = 181) | p-value | SMD |
|---|---|---|---|---|---|
| **Age (years)** | | | | 0.670 | 0.130 |
| ≥45 | 159 (17.57) | 133 (18.37) | 26 (14.36) | | |
| 46–55 | 196 (21.66) | 155 (21.41) | 41 (22.65) | | |
| 56–65 | 230 (25.41) | 186 (25.69) | 44 (24.31) | | |
| 66–75 | 256 (28.29) | 199 (27.49) | 57 (31.49) | | |
| >75 | 64 (7.07) | 51 (7.04) | 13 (7.18) | | |
| **Sex** | | | | 0.514 | 0.145 |
| Female | 331 (36.57) | 261 (36.05) | 70 (38.67) | | |
| Male | 574 (63.43) | 463 (63.95) | 111(61.33) | | |
| **ASA** | | | | 0.789 | 0.022 |
| I/II | 754 (83.31) | 602 (83.15) | 152 (83.98) | | |
| III/IV | 151 (16.69) | 122 (16.85) | 29 (16.02) | | |
| **MAP (mmHg)** | | | | 0.507 | 0.104 |
| 70–105 | 510 (56.35) | 403 (55.66) | 107 (59.12) | | |
| <70 | 11 (1.22) | 10 (1.38) | 1 (0.55) | | |
| >105 | 384 (42.43) | 311 (42.96) | 73 (40.33) | | |
| **Duration of operation (min)** | | | | 0.424 | 0.112 |
| <180 | 545 (60.22) | 431 (59.53) | 114 (62.98) | | |
| 180–300 | 211 (23.31) | 168 (23.20) | 43 (23.76) | | |
| >300 | 149 (16.46) | 125 (17.27) | 24 (13.26) | | |
| **Airway management** | | | | 0.259 | 0.168 |
| single-lumen | 29 (3.20) | 21 (2.90) | 8 (4.42) | | |
| double-lumen | 539 (59.56) | 423 (58.43) | 116 (64.09) | | |
| bronchial occluder | 328 (36.24) | 272 (37.57) | 56 (30.94) | | |
| LMA | 9 (0.99) | 8 (1.10) | 1 (0.55) | | |
| **Duration of one-lung ventilation (min)** | | | | 0.969 | 0.003 |
| ≤60 | 214 (23.65) | 171 (23.62) | 43 (23.76) | | |
| >60 | 691 (76.35) | 553 (76.38) | 138 (76.24) | | |
| **BMI (kg/m²)** | | | | 0.948 | 0.050 |
| ≤18.5 | 500 (55.25) | 401 (55.39) | 99 (54.70) | | |
| 18.5–24 | 123 (13.59) | 100 (13.81) | 23 (12.71) | | |
| 24–28 | 226 (24.97) | 178 (24.59) | 48 (26.52) | | |
| ≥28 | 56 (6.19) | 45 (6.22) | 11 (6.08) | | |
| **History of hypertension** | | | | 0.438 | 0.064 |
| No | 685 (75.69) | 552 (76.24) | 133 (73.48) | | |
| Yes | 220 (24.31) | 172 (23.76) | 48 (26.52) | | |
| **History of diabetes mellitus** | | | | 0.467 | 0.058 |
| No | 855 (94.48) | 686 (94.75) | 169 (93.37) | | |
| Yes | 50 (5.52) | 38 (5.25) | 12 (6.63) | | |

**Table 1** (*continued*)

| | Total (*n* = 905) | Training cohort (*n* = 724) | Validation cohort (*n* = 181) | *p*-value | SMD |
|---|---|---|---|---|---|
| **History of stroke** | | | | 0.151 | 0.112 |
| No | 859 (94.92) | 691 (95.44) | 168 (92.82) | | |
| Yes | 46 (5.08) | 33 (4.56) | 13 (7.18) | | |
| **History of heart disease** | | | | 0.461 | 0.063 |
| No | 833 (92.04) | 664 (91.71) | 169 (93.37) | | |
| Yes | 72 (7.96) | 60 (8.29) | 12 (6.63) | | |
| **History of COPD** | | | | 0.088 | 0.156 |
| No | 857 (94.70) | 681 (94.06) | 176 (97.24) | | |
| Yes | 48 (5.30) | 43 (5.94) | 5 (2.76) | | |
| **History of smoking** | | | | 0.938 | 0.006 |
| No | 687 (75.91) | 550 (75.97) | 137 (75.69) | | |
| Yes | 218 (24.09) | 174 (24.03) | 44 (24.31) | | |
| **History of alcohol use** | | | | 0.659 | 0.036 |
| No | 856 (94.59) | 686 (94.75) | 170 (93.92) | | |
| Yes | 49 (5.41) | 38 (5.25) | 11 (6.08) | | |
| **Leukocyte counts ($\times 10^9$/L)** | | | | 0.146 | 0.151 |
| 4-10 | 768 (84.86) | 621 (85.77) | 147 (81.22) | | |
| <4 | 99 (10.94) | 77 (10.64) | 22 (12.15) | | |
| >10 | 38 (4.20) | 26 (3.59) | 12 (6.63) | | |
| **Red blood cell counts ($\times 10^{12}$/L)** | | | | 0.321 | 0.122 |
| Normal | 750 (82.87) | 598 (82.60) | 152 (83.98) | | |
| Low | 127 (14.03) | 106 (14.64) | 21 (11.60) | | |
| High | 28 (3.09) | 20 (2.76) | 8 (4.42) | | |
| **Platelet counts ($\times 10^9$/L)** | | | | 0.273 | 0.126 |
| 100–300 | 811 (89.61) | 654 (90.33) | 157 (86.74) | | |
| <100 | 33 (3.65) | 26 (3.59) | 7 (3.87) | | |
| >300 | 61 (6.74) | 44 (6.08) | 17 (9.39) | | |
| **Aspartate aminotransferase (U/L)** | | | | 0.476 | 0.058 |
| <40 | 841 (92.93) | 675 (93.23) | 166 (91.71) | | |
| ≥40 | 64 (7.07) | 49 (6.77) | 15 (8.29) | | |
| **Alanine aminotransferase (U/L)** | | | | 0.777 | 0.049 |
| <50 | 885 (97.79) | 707 (97.65) | 178 (98.34) | | |
| ≥50 | 20 (2.21) | 17 (2.35) | 3 (1.66) | | |
| **Blood glucose (mmol/L)** | | | | 0.454 | 0.064 |
| Normal | 790 (87.29) | 629 (86.88) | 161 (88.95) | | |
| Abnormal | 115 (12.71) | 95 (13.12) | 20 (11.05) | | |
| **Blood creatinine (μmol/L)** | | | | 0.745 | 0.058 |
| <111 | 890 (98.34) | 711 (98.20) | 179 (98.90) | | |
| ≥111 | 15 (1.66) | 13 (1.80) | 2 (1.10) | | |
| **Blood urea nitrogen (mmol/L)** | | | | 0.958 | 0.027 |
| <9.5 | 882 (97.46) | 705 (97.38) | 177 (97.79) | | |
| ≥9.5 | 23 (2.54) | 19 (2.62) | 4 (2.21) | | |

**Table 1** (*continued*)

| | Total (n = 905) | Training cohort (n = 724) | Validation cohort (n = 181) | p-value | SMD |
|---|---|---|---|---|---|
| **Serum sodium (mmol/L)** | | | | 0.935 | 0.030 |
| 137–147 | 863 (95.36) | 691 (95.44) | 172 (95.03) | | |
| <137 | 25 (2.76) | 20 (2.76) | 5 (2.76) | | |
| >147 | 17 (1.88) | 13 (1.80) | 4 (2.21) | | |
| **Serum potassium (mmol/L)** | | | | 0.635 | 0.039 |
| Normal | 847 (93.59) | 679 (93.78) | 168 (92.82) | | |
| Abnormal | 58 (6.41) | 45 (6.22) | 13 (7.18) | | |
| **Serum albumin (g/L)** | | | | 0.245 | 0.097 |
| Normal | 445 (49.17) | 349 (48.20) | 96 (53.04) | | |
| Low | 460 (50.83) | 375 (51.80) | 85 (46.96) | | |
| **Types of operation** | | | | 0.510 | 0.223 |
| Trans-thoracic vagotomy | 16 (1.77) | 12 (1.66) | 4 (2.21) | | |
| Wedge resection | 111 (12.27) | 86 (11.88) | 25 (13.81) | | |
| Segmental resection | 161 (17.79) | 122 (16.85) | 39 (21.55) | | |
| Lobectomy | 295 (32.60) | 242 (33.43) | 53 (29.28) | | |
| Total pneumonectomy | 7 (0.77) | 5 (0.69) | 2 (1.10) | | |
| Partial bilateral pneumonectomy | 25 (2.76) | 22 (3.04) | 3 (1.66) | | |
| Mediastinal surgery | 63 (6.96) | 49 (6.77) | 14 (7.73) | | |
| Oesophageal surgery | 207 (22.87) | 172 (23.76) | 35 (19.34) | | |
| Others | 20 (2.21) | 14 (1.93) | 6 (3.31) | | |
| **PPCs** | | | | 0.937 | 0.007 |
| No | 698 (77.13) | 558 (77.07) | 140 (77.35) | | |
| Yes | 207 (22.87) | 166 (22.93) | 41 (22.65) | | |

**Notes.**

Values are presented as counts.

Blood biochemical data were routine examination results before the operation.

SMD, standardized mean difference; PPCs, postoperative pulmonary complications; ASA, American Society of Anesthesiologists; MAP, mean arterial pressure; LMA, laryngeal mask airway; COPD, chronic obstructive pulmonary disease.

## Development and validation of a PPC-predicting nomogram

The nomogram acquired from the training cohort to predict PPCs after thoracoscopic surgery is shown in Fig. 1. The nomogram was created based on the following 7 factors: ASA grade (I/II or III/IV), operation time (<180, 180–300, or ≥ 300 min), one-lung ventilation time (<60 or ≥ 60 min), history of stroke (no or yes), history of heart disease (no or yes), history of COPD (no or yes), and history of smoking (no or yes). A higher point total based on the sum of the assigned number of points for each factor in the nomogram was associated with a higher risk of PPCs.

The total scores of the nomogram of each patient in the training cohort were calculated. ROC curves were analyzed to calculate the optimal cut-off point based on the Youden index in the training cohort, which yielded a value of 134 points. The accuracy, precision, sensitivity, specificity, and C-index are presented in Table 3. This cut-off point was used in both cohorts. The areas under the ROC curve (AUCs) were 0.894 (95% CI [0.866–0.922]) and 0.867 (95% CI [0.810–0.925]) in the training and validation cohorts, respectively

**Table 2  Univariate and multivariable logistic regression analyses of variables for PPCs after thoracoscopic surgery in the training cohort.**

| Project | | Univariate analysis | | Multivariate analysis | |
|---|---|---|---|---|---|
| | | OR (95% CI) | *p*-value | OR (95% CI) | *p*-value |
| Age (years) | ≤45 | Ref | | | |
| | 46–55 | 2.515 (1.066–6.622) | 0.045 | | |
| | 56–65 | 5.250 (2.416–13.155) | <0.001 | | |
| | 66–75 | 10.428 (4.930–25.704) | <0.001 | | |
| | >75 | 17.309 (7.097–47.375) | <0.001 | | |
| Sex | female | Ref | | | |
| | male | 1.782 (1.215–2.653) | 0.004 | | |
| ASA | I/II | Ref | | Ref | |
| | III/IV | 14.776 (9.485–23.437) | <0.001 | 10.171 (5.598–18.926) | <0.001 |
| MAP (mmHg) | 70–105 | Ref | | | |
| | <70 | 0.404 (0.022–2.190) | 0.392 | | |
| | >105 | 1.216 (0.857–1.724) | 0.273 | | |
| Duration of operation (min) | <180 | Ref | | Ref | |
| | 180–300 | 3.229 (2.073–5.037) | <0.001 | 2.480 (1.363–4.563) | 0.003 |
| | >300 | 8.524 (5.414–13.563) | <0.001 | 12.516 (5.232–17.766) | <0.001 |
| Airway management | Bronchial occluder | Ref | | | |
| | double-lumen | 1.335 (0.438–5.800) | 0.650 | | |
| | LMA | 2.774 (0.909–12.067) | 0.109 | | |
| | single-lumen | 0.000 (0.000–0.000) | 0.979 | | |
| Duration of one-lung ventilation (min) | ≤60 | Ref | | Ref | |
| | >60 | 1.000 (1.000–1.000) | <0.001 | 2.947 (1.257–7.648) | 0.018 |
| BMI (kg/m$^2$) | <18.5 | Ref | | | |
| | 18.5–24 | 0.625 (0.344–1.080) | 0.105 | | |
| | 24–28 | 0.942 (0.619–1.417) | 0.777 | | |
| | ≥28 | 1.516 (0.277–1.395) | 0.306 | | |
| History of hypertension | No | Ref | | | |
| | Yes | 1.481 (0.999–2.177) | 0.048 | | |
| History of diabetes mellitus | No | Ref | | | |
| | Yes | 1.812 (0.880–3.567) | 0.093 | | |
| History of stroke | No | Ref | | Ref | |
| | Yes | 5.022 (2.472–10.442) | <0.001 | 4.209 (1.518–11.683) | 0.006 |
| History of heart disease | No | Ref | | Ref | |
| | Yes | 7.233 (4.171–12.832) | <0.001 | 2.001 (0.901–4.479) | 0.089 |
| History of COPD | No | Ref | | Ref | |
| | Yes | 7.345 (3.871–14.467) | <0.001 | 2.888 (1.230–6.980) | 0.016 |
| History of smoking | No | Ref | | Ref | |
| | Yes | 5.962 (4.083–8.753) | <0.001 | 4.101 (2.544–6.654) | <0.001 |
| History of alcohol use | No | Ref | | | |
| | Yes | 1.594 (0.760–3.168) | 0.196 | | |

**Table 2** (*continued*)

| Project | | Univariate analysis | | Multivariate analysis | |
|---|---|---|---|---|---|
| | | OR (95% CI) | *p*-value | OR (95% CI) | *p*-value |
| **Leukocyte counts** ($\times 10^9$/L) | 4–10 | Ref | | | |
| | <4 | 1.325 (0.760–2.234) | 0.304 | | |
| | >10 | 1.570 (0.633–3.577) | 0.301 | | |
| **Red blood cell counts** ($\times 10^{12}$/L) | Normal | Ref | | | |
| | Low | 1.828 (1.157–2.850) | 0.009 | | |
| | Hight | 0.927 (0.262–2.580) | 0.894 | | |
| **Platelet counts** ($\times 10^9$/L) | 100–300 | Ref | | | |
| | <100 | 1.227 (0.472–2.852) | 0.650 | | |
| | >300 | 0.740 (0.314–1.548) | 0.454 | | |
| **Aspartate aminotransferase (U/L)** | <40 | Ref | | | |
| | ≥40 | 0.971 (0.463–1.883) | 0.934 | | |
| **Alanine aminotransferase (U/L)** | <50 | Ref | | | |
| | ≥50 | 1.413 (0.444–3.871) | 0.522 | | |
| **Blood glucose (mmol/L)** | Normal | Ref | | | |
| | Abnormal | 0.882 (0.509–1.468) | 0.641 | | |
| **Blood creatinine ($\mu$mol/L)** | <111 | Ref | | | |
| | ≥111 | 2.135 (0.638–6.489) | 0.189 | | |
| **Blood urea nitrogen (mmol/L)** | <9.5 | Ref | | | |
| | ≥9.5 | 1.572 (0.545–4.047) | 0.367 | | |
| **Serum sodium (mmol/L)** | 137–147 | Ref | | | |
| | <137 | 1.143 (0.367–3.003) | 0.799 | | |
| | >147 | 2.143 (0.640–6.518) | 0.187 | | |
| **Serum potassium (mmol/L)** | Normal | Ref | | | |
| | Abnormal | 1.566 (0.790–2.964) | 0.181 | | |
| **Serum albumin (g/L)** | Normal | Ref | | | |
| | Low | 1.611 (1.134–2.300) | 0.008 | | |
| **Types of operation** | Wege resection | Ref | | | |
| | Trans-thoracic vagotomy | 0.000 (0.000–0.000) | 0.983 | | |
| | Segmental resection | 1.180 (0.533–2.717) | 0.687 | | |
| | Lobectomy | 1.687 (0.859–3.578) | 0.147 | | |
| | Total pneumonectomy | 0.000 (0.000,inf) | 0.989 | | |
| | Partial bilateral pneumonectomy | 2.005 (0.571–6.333) | 0.248 | | |
| | Mediastinal surgery | 0.290 (0.044–1.142) | 0.118 | | |
| | Esophageal surgery | 6.069 (3.122–12.808) | <0.001 | | |
| | Others | 0.524 (0.028–3.062) | 0.553 | | |

**Notes.**

Blood laboratory data were routine examination results before the operation.

OR, odds ratio; CI, confidence interval; ASA, American Society of Anesthesiologists; MAP, mean arterial pressure; LMA, laryngeal mask airway; COPD, chronic obstructive pulmonary disease.

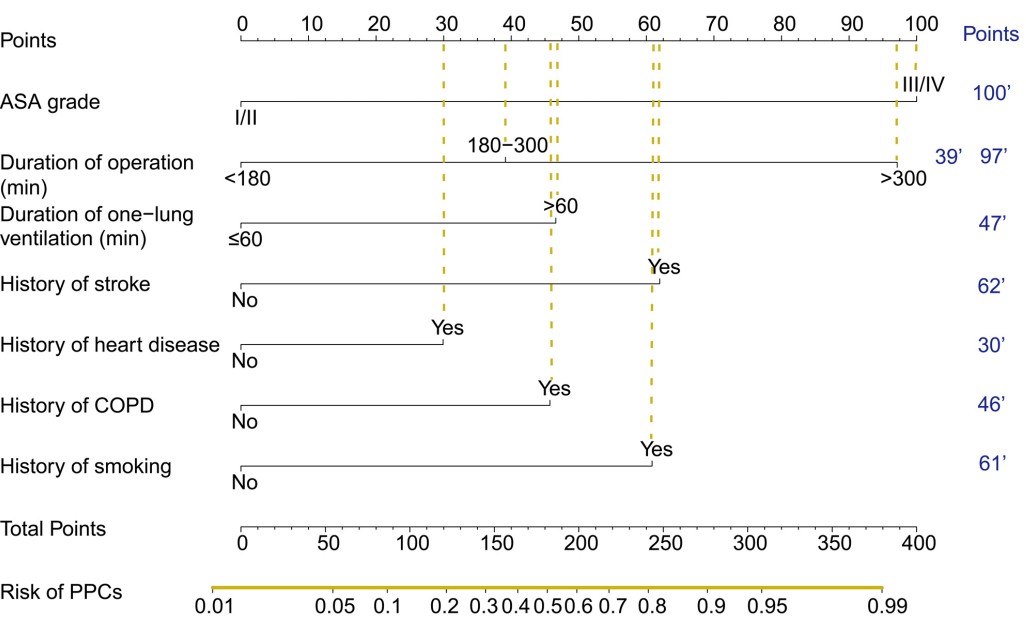

**Figure 1** The nomogram used to predict PPCs in patients following thoracoscopic surgery.

**Table 3** Prediction performance of the nomogram in the training cohort and the validation cohort.

|  | Training cohort (*n* = 724) | Validation cohort (*n* = 181) |
| --- | --- | --- |
| **PPCs** | 166 (22.93%) | 41 (22.65%) |
| **Nomogram points (Mean ±SD)** | 101.27 ± 84.78 | 96.55 ± 78.54 |
| **Nomogram points (Min, Max)** | (0, 397) | (0, 305) |
| **cut-off points** | 134 | 134 |
| **Accuracy** | 0.817 | 0.807 |
| **Precision** | 0.567 | 0.552 |
| **Sensitivity** | 0.855 | 0.814 |
| **Specificity** | 0.806 | 0.780 |
| **AUC of nomogram (95% CI)** | 0.894(0.866–0.922) | 0.867(0.810–0.925) |
| **C-index of nomogram (95% CI)** | 0.894(0.866–0.921) | 0.868(0.811–0.925) |

**Notes.**
The nomogram generated by the training cohort.
PPCs, postoperative pulmonary complications; SD, standard deviation; AUC, area under the receiver operating characteristic curve; CI, confidence interval.

(Fig. 2). In addition, the C-indices with 1000 bootstrap replicates were 0.894 (95% CI [0.866–0.921]) and 0.868 (0.811−0.925) in the training and validation cohorts, respectively (Table 2), indicating the good performance of the nomogram in estimating the risk of PPCs. The calibration plots with 100 bootstrap results of the training and validation cohorts are shown in Fig. 3, which vividly demonstrates good agreement regarding the presence of PPCs between the risk estimation by the nomogram acquired from the training cohort and the actual PPC occurrence in the training and validation cohorts.
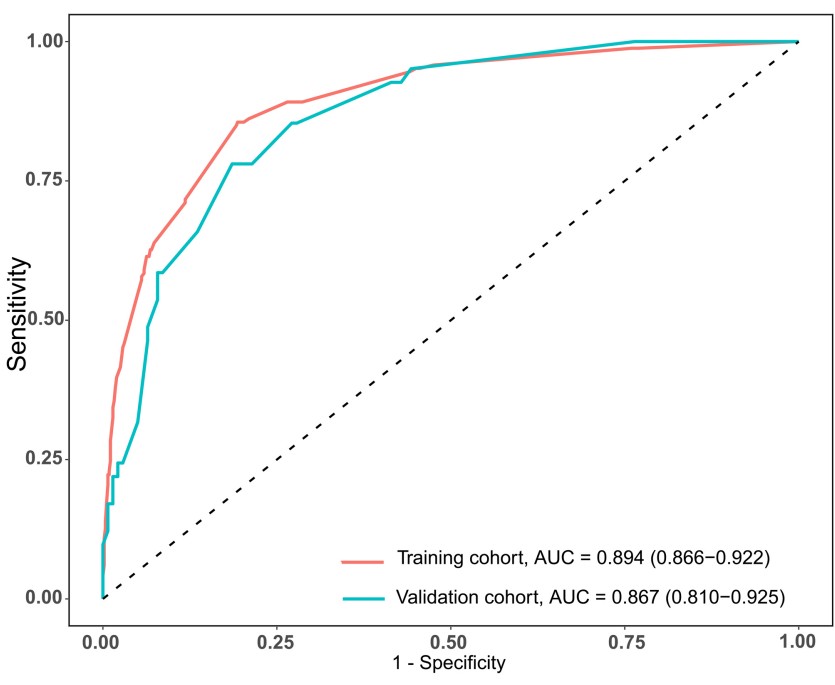

**Figure 2** AUCs of the nomogram used to predict PPCs following thoracoscopic surgery in the training and validation cohorts.

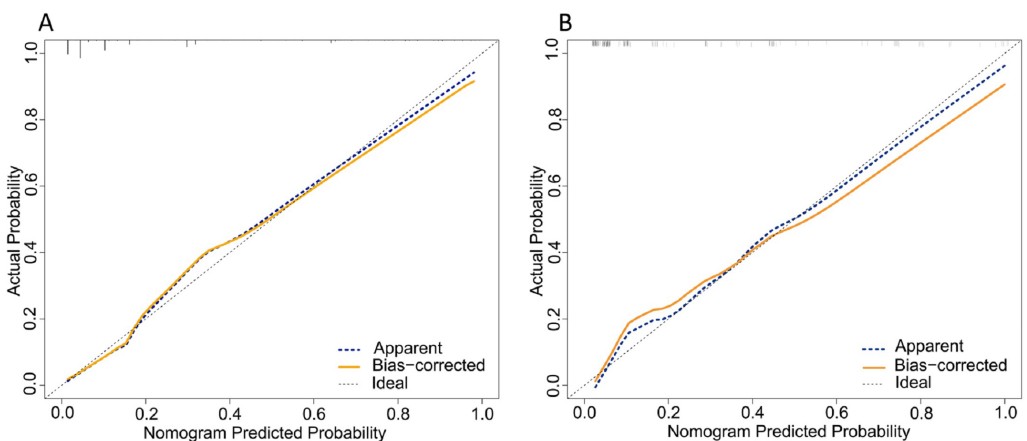

**Figure 3** Calibration plot comparing predicted and actual PPCs following thoracoscopic surgery in the training and validation cohorts.

## DISCUSSION

In the current study, a nomogram was developed to predict the risk of PPCs after thoracoscopic surgery. This nomogram was constructed using the training cohort and achieved good performance in both the training and validation cohorts; the ROC and C-index results were also good in both cohorts. Nomogram is a very useful tool to predict

diseases or outcomes (*Gu et al., 2020*; *Kong et al., 2019*; *Lei et al., 2016*; *Li & Meng, 2019*; *Lu et al., 2019*). The nomogram constructed in this study had good predictive performance and generalizability. Overall, building a prediction model for PPCs after thoracoscopic surgery is important for the perioperative management of patients, and the model can help with early intervention for patient complications, thus improving the quality of surgery and reducing medical costs. Moreover, obtaining information about patient risk factors, such as medical history or routine preoperative examinations, is conducive to appropriate clinical practice and implementation.

In this study, the incidence of PPCs during thoracoscopic surgery was 22.9%. In general, thoracic surgery has a high incidence of PPCs. Although thoracoscopic surgery reduces the incidence, PPCs are still common. The incidence of PPCs in this study was consistent with previous research results (*Kaufmann et al., 2019*; *Nozaki et al., 2018*) but higher than that in some studies (*Agostini et al., 2017*; *Ceppa et al., 2012*). In the present study, the diagnostic criteria for PPCs were based on EPCO guidelines (*Jammer et al., 2015*; *Miskovic & Lumb, 2017*), which were published in 2015 to standardize the definition of PPCs, as these are composite outcome indicators and exhibit certain differences in clinical evaluation criteria. Respiratory infection, respiratory failure, pleural effusion, atelectasis, pneumothorax, bronchospasm, and aspiration pneumonia are included in the guidelines that define PPCs. However, the incidence of PPCs in this study was somewhat high because of the broad definition of PPCs in the guidelines, *e.g.*, pure pleural effusion is also listed as a pulmonary complication.

The variables incorporated into the nomogram are easy to obtain clinically, rendering the model easy to extend and apply, and seven indicators were included: ASA grade, duration of operation time, duration of one-lung ventilation, and history of stroke, heart disease, COPD, and smoking. Nomograms are simple and practical and can be widely used in clinical practice with high applicability (*Jung et al., 2019*; *Lei et al., 2016*; *Yap et al., 2018*; *Yuan et al., 2020*). When using the nomogram, only one variable should be positioned on the corresponding axis, and the points of all variables are summed to determine risk. Some scholars have carried out nomogram research in the field of anesthesia and achieved good results (*Ferrando et al., 2020*; *Ge et al., 2018*; *Hahn, 1995*; *Zhou et al., 2019*).

The current study involved patients who had undergone thoracoscopic surgery, and the independent risk factors were similar to those in a previous study. Indeed, the reported risk factors for pulmonary complications in thoracic surgery include older age, cardiovascular complications, smoking, COPD history, and high BMI and ASA grade (*Agostini et al., 2010*; *Licker et al., 2006*; *Stephan et al., 2000*). By using backward stepwise selection with the AIC in logistic regression modelling, we identified the following 7 variables as having the strongest associations with PPC risk: ASA grade III/IV, operation time longer than 180 min, one-lung ventilation time longer than 60 min, and history of stroke, heart disease, COPD, and smoking. Conversely, some indicators, such as older age, were not included as predictors of risk. In the univariate logistic regression, age differed significantly, but it was not an independent risk factor in the multivariate regression. The reason for this finding may be that elderly patients with other diseases, such as stroke, COPD, and heart disease,

in our study population had higher ASA grades. Nonetheless, older age and these factors have obvious collinearity.

The nomogram could be used to predict the risk of PPCs in thoracoscopic surgery. Higher the total points obtained from each indicator in the nomogram means greater the possibility of PPCs. Because of the convenience to obtain the indicators in clinical work, so it is convenient to evaluate whether patients are prone to postoperative complications according to the nomogram, so as to early identify high-risk patients, and early detect, intervene, and improve postoperative outcomes of patients. For example, an ASA grade III patient undergoing thoracoscopic surgery with 200 min (>180 min) operation time and 80 min one-lung ventilation time (>60 min), with history of COPD and smoking while without history of stroke or heart disease would have a total of 293 points for PPCs (100 points for ASA grade III, 39 points for duration of operation, 47 points for duration of one-lung ventilation, 0 points for history of stroke, 0 points for history of heart disease, 46 points for history of COPD, and 61 points for history of smoking). The predicted PPCs risk of this patient was about 92%.

The present study has some limitations. This was a retrospective study, and PPCs may be missed or misdiagnosed. The case data may not fully reflect the PPCs and diseases of patients, and some data could not be obtained. Moreover, this was a single-center study with a small sample size, and all thoracoscopic surgeries were included. Some variables, such as blood transfusion, could not be included in the model because of too few transfusion patients and large random errors. PPCs have a relatively broad and general definition, and the degree of specific complications was not analyzed. Multicenter, large-sample, prospective studies and subgroup analyses should be carried out in the future.

## CONCLUSIONS

A nomogram was constructed by combining seven risk factors for PPCs. The model provides a good estimation of PPC risk in patients after thoracoscopic surgery, which may be useful for patients who undergo thoracoscopic surgery.

**Abbreviations**

| | |
|---|---|
| **PPC** | Postoperative pulmonary complication |
| **AIC** | Akaike information criterion |
| **ASA** | American Society of Anesthesiologists |
| **COPD** | chronic obstructive pulmonary disease |
| **CI** | confidence interval |
| **BMI** | body mass index |
| **AIDS** | acquired immunodeficiency syndrome |
| **MAP** | mean arterial pressure |
| **EPCO** | European Perioperative Clinical Outcome |
| **ROC** | receiver operating characteristic |
| **SMD** | standardized mean difference |
| **OR** | odds ratio |
| **AUC** | area under the ROC curve |
| **LMA** | laryngeal mask airway |
| **SD** | standard deviation |

## ACKNOWLEDGEMENTS

We thank Dr Hui Peng and Dr Lin Sun for their help with the statistical analysis. We also thank the patients who participated in this study.

### Funding

This study was supported by the Department of Science and Technology of Anhui Province, Hefei, China (No. 1704a0802165) and the Natural Science Foundation of Anhui Province (No. 1908085QH358). The funders had no role in study design, data collection and analysis, decision to publish, or preparation of the manuscript.

### Grant Disclosures

The following grant information was disclosed by the authors:
The Department of Science and Technology of Anhui Province, Hefei, China: 1704a0802165.
The Natural Science Foundation of Anhui Province: 1908085QH358.

### Competing Interests

The authors declare there are no competing interests.

### Author Contributions

- Bin Wang conceived and designed the experiments, performed the experiments, analyzed the data, prepared figures and/or tables, authored or reviewed drafts of the paper, and approved the final draft.
- Zhenxing Chen performed the experiments, prepared figures and/or tables, authored or reviewed drafts of the paper, and approved the final draft.
- Ru Zhao performed the experiments, analyzed the data, authored or reviewed drafts of the paper, and approved the final draft.
- Li Zhang performed the experiments, analyzed the data, prepared figures and/or tables, and approved the final draft.
- Ye Zhang conceived and designed the experiments, analyzed the data, prepared figures and/or tables, authored or reviewed drafts of the paper, and approved the final draft.

### Human Ethics

The following information was supplied relating to ethical approvals (i.e., approving body and any reference numbers):
    This study was approved by the Ethics Committee of the Second Affiliated Hospital of Anhui Medical University.

### Clinical Trial Ethics

The following information was supplied relating to ethical approvals (i.e., approving body and any reference numbers):
    This study was approved by the Ethics Committee of the Second Affiliated Hospital of Anhui Medical University.
## Data Availability

The raw data is available in the Supplementary Files.

## Clinical Trial Registration

The following information was supplied regarding Clinical Trial registration:

This study was registered in the Chinese Clinical Trial Registry (No. ChiCTR2000033329).

## Supplemental Information

Supplemental information for this article can be found online at http://dx.doi.org/10.7717/peerj.12366#supplemental-information.

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
