# Peer review of "Development and validation of a nomogram to predict postoperative pulmonary complications following thoracoscopic surgery"

_PeerJ, doi:10.7717/peerj.12366_

## Round 0.1 · original submission · Major Revisions

The manuscript you submitted to PeerJ has been reviewed. The reviewers have recommended publication pending major revisions.

In addition to the comments from the reviewers, please also address the following:

(1) The surgical approaches, such as lobectomy, sub-lobectomy, and pneumonectomy, as well as lymph nodes dissection, may also affect the probability of postoperative pulmonary complications. The authors should better include the surgical approaches in their univariate logistic regression analyses.

(2) Please give the details about the definition of postoperative pulmonary complications, especially how long it occurs since the surgery.

I invite you to respond to these, and the reviewers' comments at the bottom of this letter and revise your manuscript accordingly.

Reviewer 1 ·

Basic reporting

No comment

Experimental design

No comment

Validity of the findings

No comment

Additional comments

I read with interest the manuscript by Wang et al, and appreciate the opportunity to provide feedback. Here are my comments:

1) Do you have the data regarding blood loss volume or the use of transfusion during operation?
2) Although authors mentioned about the relation between thoracoscopic surgery and postoperative complications, they did not compare thoracoscopic surgery with thoracotomy.
3) This study cohort included various types of surgery such as mediastinal and esophageal surgery. I think that the extent of lung resection could greatly influence on the incident of pulmonary complications. Therefore, authors should examine the surgical types in univariate and multivariate analysis for PPC.

·

Basic reporting

No comment. Good clear reporting

Experimental design

Well defined objectives

Validity of the findings

To elaborate how will this normogram impact clinical care

Additional comments

nil

---

## Round 0.2 · accepted · Accept

This manuscript has been greatly improved.

Reviewer 1 ·

Basic reporting

The changes made are appreciated and have improved the paper. This study should be of some interest to readers.

Experimental design

no concerns

Validity of the findings

no concern